# Orbital period change of Dimorphos due to the DART kinetic impact

Cristina A. Thomas[1✉], Shantanu P. Naidu[2], Peter Scheirich[3], Nicholas A. Moskovitz[4], Petr Pravec[3], Steven R. Chesley[2], Andrew S. Rivkin[5], David J. Osip[6], Tim A. Lister[7], Lance A. M. Benner[2], Marina Brozović[2], Carlos Contreras[6], Nidia Morrell[6], Agata Rożek[8], Peter Kušnirák[3], Kamil Hornoch[3], Declan Mages[2], Patrick A. Taylor[9], Andrew D. Seymour[10], Colin Snodgrass[8], Uffe G. Jørgensen[11], Martin Dominik[12], Brian Skiff[4], Tom Polakis[4], Matthew M. Knight[13], Tony L. Farnham[14], Jon D. Giorgini[2], Brian Rush[2], Julie Bellerose[2], Pedro Salas[10], William P. Armentrout[10], Galen Watts[10], Michael W. Busch[15], Joseph Chatelain[7], Edward Gomez[7,16], Sarah Greenstreet[17], Liz Phillips[7,18], Mariangela Bonavita[8], Martin J. Burgdorf[19], Elahe Khalouei[20], Penélope Longa-Peña[21], Markus Rabus[22], Sedighe Sajadian[23], Nancy L. Chabot[5], Andrew F. Cheng[5], William H. Ryan[24], Eileen V. Ryan[24], Carrie E. Holt[14] & Harrison F. Agrusa[14]

The Double Asteroid Redirection Test (DART) spacecraft successfully performed the first test of a kinetic impactor for asteroid deflection by impacting Dimorphos, the secondary of near-Earth binary asteroid (65803) Didymos, and changing the orbital period of Dimorphos. A change in orbital period of approximately 7 min was expected if the incident momentum from the DART spacecraft was directly transferred to the asteroid target in a perfectly inelastic collision[1], but studies of the probable impact conditions and asteroid properties indicated that a considerable momentum enhancement ($\beta$) was possible[2,3]. In the years before impact, we used lightcurve observations to accurately determine the pre-impact orbit parameters of Dimorphos with respect to Didymos[4–6]. Here we report the change in the orbital period of Dimorphos as a result of the DART kinetic impact to be −33.0 ± 1.0 (3$\sigma$) min. Using new Earth-based lightcurve and radar observations, two independent approaches determined identical values for the change in the orbital period. This large orbit period change suggests that ejecta contributed a substantial amount of momentum to the asteroid beyond what the DART spacecraft carried.

NASA's DART successfully impacted Dimorphos, the secondary of the near-Earth binary asteroid (65803) Didymos, on 26 September 2022 at 23:14 UTC. The primary objective of DART was to change the orbital period of Dimorphos around Didymos to demonstrate that a kinetic impactor is a viable method of asteroid deflection[1,7]. The mission targeted the secondary asteroid in an eclipsing binary system because the experiment could use a single impacting spacecraft and measure the change in the orbit of the secondary through ground-based observations. The Didymos system was selected as the target because it is among the most accessible (low $\Delta V$) of the near-Earth binaries, it has been extremely well characterized[4–6,8–12] and Dimorphos is in the size range identified as relevant for deflection by a kinetic impactor[13,14].

The DART spacecraft collided head-on into the leading hemisphere of Dimorphos to maximize the momentum transfer and reduce the semimajor axis of the Dimorphos orbit, resulting in a shorter orbital period[7]. If the incident momentum from the impacting spacecraft was simply transferred to the asteroid target with no further momentum enhancement, an orbital period change for Dimorphos of roughly 7 min was expected[1]. Impact simulations conducted in preparation for DART's kinetic impact test indicated that, depending on the material strength, impact conditions and other properties, the value of the momentum enhancement factor, $\beta$, could be considerable, with predicted values as high as 5 (ref. 2) or 6 (ref. 3), with a resulting orbital period change of more than 40 min (ref. 15).

The Didymos system lightcurve is composed of three parts: the rotational lightcurve of Didymos, the rotational lightcurve of Dimorphos and the mutual events that constrain the orbital period. The Didymos rotational lightcurve can be clearly distinguished because the primary contributes approximately 96% of the light from the system. The Dimorphos rotational period has not been resolved because of

[1]Northern Arizona University, Flagstaff, AZ, USA. [2]Jet Propulsion Laboratory, California Institute of Technology, Pasadena, CA, USA. [3]Astronomical Institute of the Czech Academy of Sciences, Ondřejov, Czech Republic. [4]Lowell Observatory, Flagstaff, AZ, USA. [5]Johns Hopkins University Applied Physics Laboratory, Laurel, MD, USA. [6]Carnegie Institution for Science, Las Campanas Observatory, La Serena, Chile. [7]Las Cumbres Observatory, Goleta, CA, USA. [8]University of Edinburgh, Royal Observatory, Edinburgh, UK. [9]National Radio Astronomy Observatory, Charlottesville, VA, USA. [10]Green Bank Observatory, Green Bank, WV, USA. [11]Niels Bohr Institute, University of Copenhagen, Copenhagen, Denmark. [12]University of St Andrews, St Andrews, UK. [13]United States Naval Academy, Annapolis, MD, USA. [14]University of Maryland, College Park, MD, USA. [15]SETI Institute, Mountain View, CA, USA. [16]Cardiff University, Cardiff, UK. [17]University of Washington, Seattle, WA, USA. [18]University of California, Santa Barbara, Santa Barbara, CA, USA. [19]Universität Hamburg, Hamburg, Germany. [20]Seoul National University, Gwanak-gu, Seoul, Korea. [21]Universidad de Antofagasta, Antofagasta, Chile. [22]Universidad Católica de la Santísima Concepción, Concepción, Chile. [23]Isfahan University of Technology, Isfahan, Iran. [24]Magdalena Ridge Observatory, New Mexico Institute of Mining and Technology, Socorro, NM, USA. ✉e-mail: cristina.thomas@nau.edu

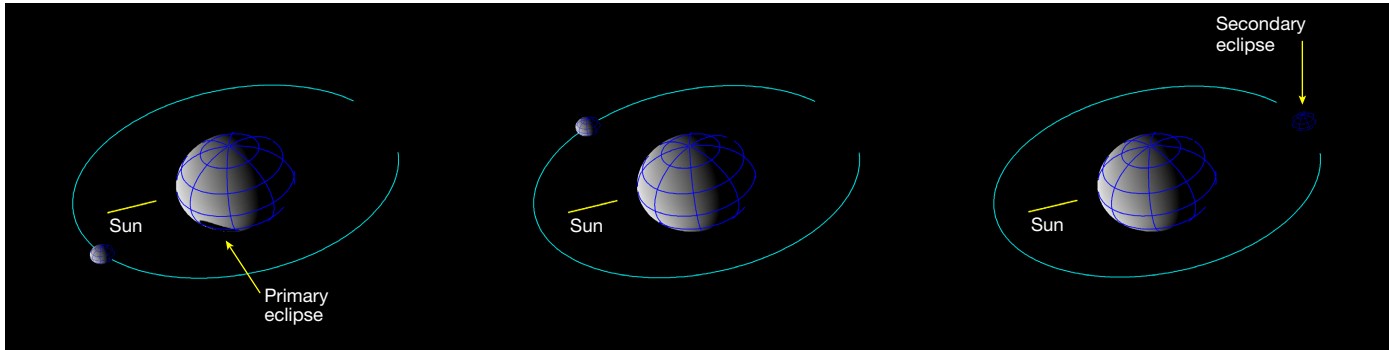

**Fig. 1 | Post-impact Didymos system geometry.** We determine the new orbital period of Dimorphos using the times of mutual events, when a measurable decrease in the system brightness occurs because of an eclipse or occultation. Due to the geometry of the Didymos system during this time period, our lightcurve observations include primary eclipses (left), time outside mutual events (centre) and secondary eclipses (right). These diagrams simulate the view of the system from Earth on 10 October 06:09 (primary eclipse), 10 October 08:47 (outside events) and 10 October 12:06 (secondary eclipse) in geocentric UTC. The primary eclipses observed throughout our post-impact dataset are grazing, which resulted in a subtle decrease in system brightness (Fig. 3). During the secondary eclipse, Dimorphos is completely shadowed.

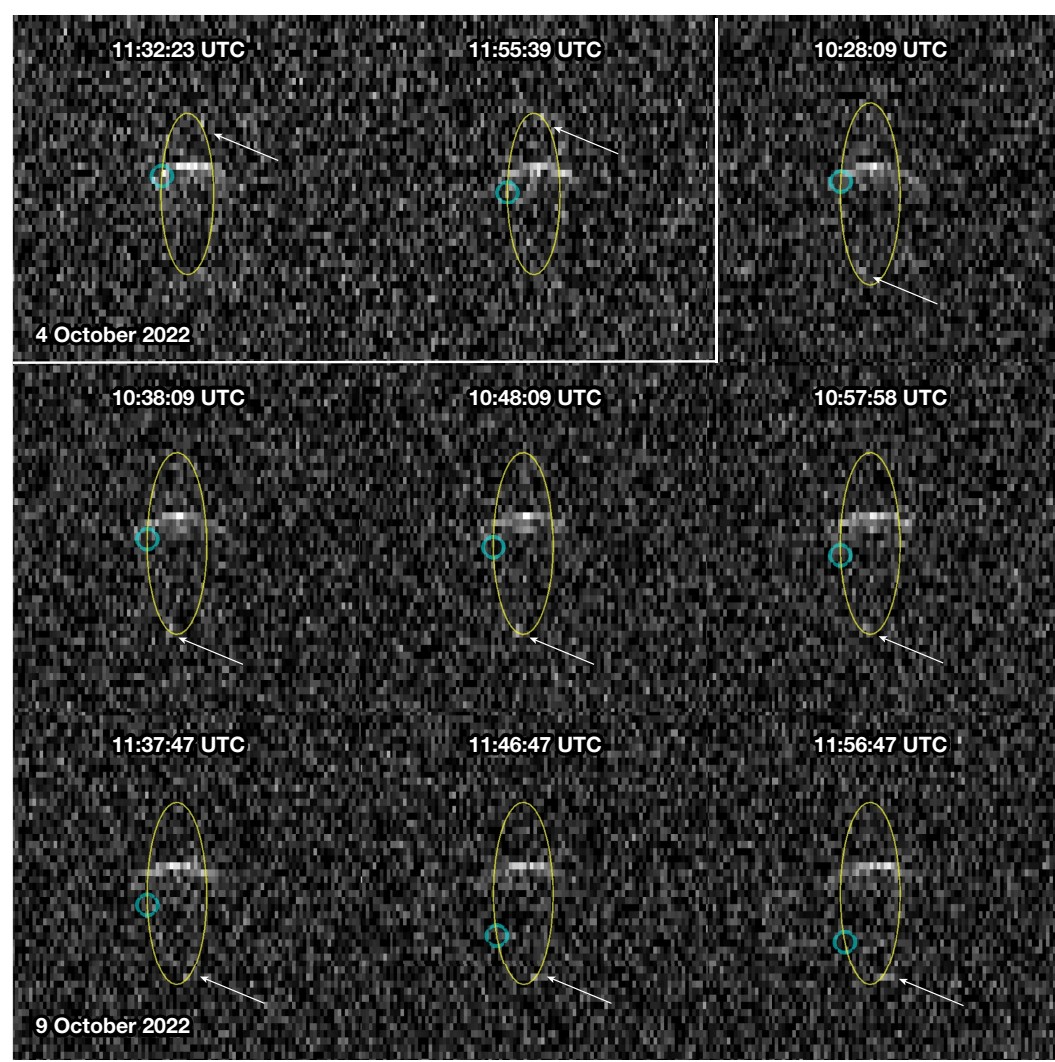

**Fig. 2 | Radar range-Doppler images of the post-impact Didymos system.** Radar range-Doppler images obtained on 4 October using Goldstone and 9 October using Goldstone to transmit and the Green Bank Telescope to receive. In each image, the distance from Earth increases from top to bottom and the Doppler frequency increases to the right, so rotation and orbital motion are anticlockwise. Each image was integrated for 20 min, with 10 min of overlap between successive images. Images have resolutions of 75 m × 0.5 Hz. The broader echo is from Didymos and the smaller, fainter echo shown using arrows is from Dimorphos. The open circles show Dimorphos positions predicted by the pre-impact orbit. The yellow ellipses show the trajectory of Dimorphos. Prediction uncertainties are smaller than the image resolution. On 4 October, the ellipse spans −870 m to +870 m along the $y$ axis and −7 Hz to +7 Hz along the $x$ axis, corresponding to line-of-sight velocity of −12 cm s$^{-1}$ to +12 cm s$^{-1}$. On 9 October, the ellipse spans −980 m to +980 m along the $y$ axis and −8 Hz to +8 Hz along the $x$ axis, corresponding to line-of-sight velocity of −14 cm s$^{-1}$ to +14 cm s$^{-1}$. The physical extents of the ellipse vary because of the viewing geometry.

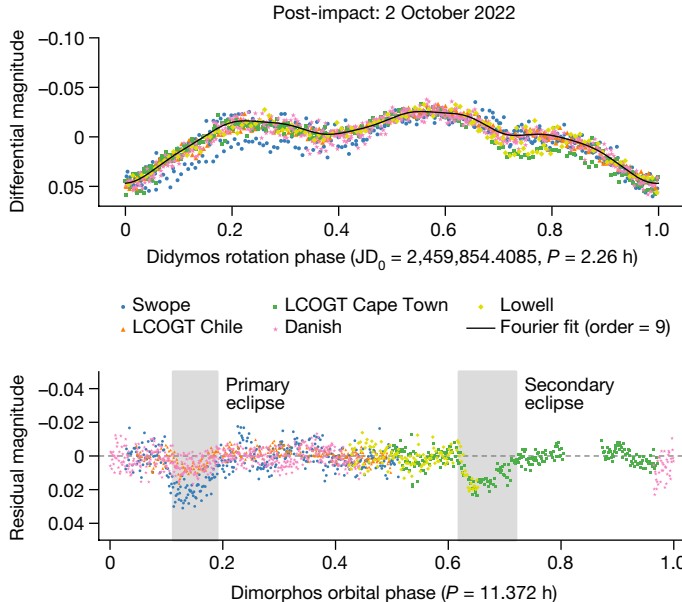

**Post-impact: 2 October 2022**

Didymos rotation phase (JD$_0$ = 2,459,854.4085, P = 2.26 h)

- Swope
- LCOGT Chile
- LCOGT Cape Town
- Danish
- Lowell
- — Fourier fit (order = 9)

Primary eclipse    Secondary eclipse

Dimorphos orbital phase (P = 11.372 h)

**Fig. 3 | Post-impact photometric lightcurve of the Didymos system.** Measured photometry from UTC 2 October 2022 phase folded to the 2.26-h rotation period of Didymos (top) and the extracted mutual events (= observed data − 9th order Fourier fit to the rotation of Didymos) phase folded to the new orbit period of Dimorphos (bottom). These lightcurves, collected from five different telescopes, show photometric accuracy similar to all the lightcurve datasets in our analysis. The mutual event times are highly consistent across these datasets, although residual systematics in the photometry result in slightly different event depths.

its comparatively small size, the oblate shape of Dimorphos[16] and the accuracy of the photometric observations necessary for such a detection. Mutual events cause a measurable decrease in the total brightness of the system. We define the primary/secondary occultation or eclipse based on which object is being obscured or shadowed, respectively. We use the timings of the observed mutual events in the determination of the orbital period. For the Didymos–Dimorphos system, mutual events occur when the Didymos–Sun or the Didymos–Earth vector forms an angle less than approximately 17° with the mutual orbit plane of the system. Since the inclination of the mutual orbit to the heliocentric orbit of the binary system is lower than this value, eclipses (mutual shadowing of the components; Fig. 1) always occur. Occultations did not occur during the observing period presented in this paper.

A precise determination of the pre-impact orbital parameters of the Didymos system was a key goal once the system was chosen as the target of DART. The initial orbit of Dimorphos was first defined following the 2003 apparition when the secondary was discovered[11,17]. Analyses of lightcurve-derived mutual events obtained during 2003–2022 (ref. 4) led to independent and consistent orbital periods[5,6]. The data used in the published pre-impact orbit solutions were augmented with photometric data obtained in July 2022 to calculate the pre-impact orbit period for Dimorphos (Extended Data Table 1). Both approaches determined a statistically identical pre-impact orbital period of 11.92148 ± 0.00013 (3σ) h.

To determine the post-impact orbital period, we obtained radar and lightcurve observations of the Didymos system. Our radar observations of Didymos and Dimorphos began about 11 h after impact using the Goldstone X-band (3.5 cm, 8,560 MHz) and continued for 14 dates between UTC 27 September and 13 October (all subsequent dates are in UTC). We also used the Green Bank Telescope to receive radar echoes in a bistatic configuration with transmissions from Goldstone on 2, 6 and 9 October. We obtained echo power spectra during each of the observing windows and range-Doppler images (Fig. 2) on 10 days centred on

4 October, when the signal-to-noise ratios (SNRs) were the highest because Didymos was the closest to Earth. The radar observations of the system are not subject to the same shadowing geometry as the lightcurve photometry. Dimorphos can be seen when illuminated by radar and the system was never in a radar eclipse geometry. We measured the separations between Dimorphos and Didymos in the echo power spectra and the range-Doppler images. We used these measurements in the determination of the orbital parameters of Dimorphos relative to Didymos. We only used data in which the SNRs were strong enough to detect both Didymos and Dimorphos. The first observation of Dimorphos (8σ detection), approximately 12 h after impact, yielded the first estimate of the orbital period change of −36 ± 15 min.

Following the DART kinetic impact, ejecta was introduced into the system[18]. The extra flux and the variable brightness from the rapidly evolving ejecta prevented immediate observations of the mutual events. Lightcurve observations began in the hours after impact and our first successful detection of a mutual event was a secondary eclipse approximately 29.5 h after impact (mid-time at geocentric UTC 28 September 04:50). At the time of the first mutual event detection, the flux from the ejecta dominated the signal within the photometric aperture. This contamination resulted in a reduction in the observed amplitude of the Didymos rotational lightcurve by a factor of 3. The apparent depth of the secondary eclipse was also markedly reduced compared with the predictions[6]. Pre-impact ejecta models[19] suggested that it could take up to several days for our ground-based lightcurve observations to detect the first mutual event as a result of the total ejecta brightness and because the rate of change of that brightness could be comparable with the expected changes in the Didymos system brightness during mutual events.

Photometric observations included in this analysis were obtained from 28 September to 10 October 2022 (Extended Data Table 2). This set of observations ends on 10 October because subsequent observations did not have the required precision because of the bright Moon. On average, our data have photometric accuracy of root mean square (RMS) about 0.006 magnitudes. The exceptional quality of the data included in our analysis has enabled the determination of the Dimorphos orbital period change from lightcurves despite the presence of ejecta in all of our observations (Figs. 3 and 4). At the time of these first observations, the primary eclipses were grazing events (Fig. 1), which required exceptionally precise data to measure.

Two independent methods were used to model the available data for determination of the post-impact orbital period: (1) we use the processes described in ref. 6 to model the lightcurve observations alone and (2) we combine the radar and mutual event timings[5,11] plus Didymos-relative astrometry of Dimorphos in optical navigation images from the Didymos Reconnaissance and Asteroid Camera for Optical navigation (DRACO) on the DART spacecraft[20]. Both methods use the same ground-based photometric datasets but have independent processes for accepting individual data points and mutual events. Ellipsoidal approximations of the shapes of Dimorphos and Didymos are incorporated in the calculation of the orbit period of Dimorphos in both approaches and the axial ratios reported in ref. 16 were used for their calculation.

We determine a post-impact period of 11.372 ± 0.017 (3σ) h with a period change of −33.0 ± 1.0 (3σ) min. Both methods provide statistically identical results for the post-impact orbital period. The rotation period of Didymos is measured during the lightcurve analysis process and shows no variation from its pre-impact value of 2.260 h to an uncertainty of approximately 5 s (3σ). The rotational lightcurve of Dimorphos has not been detected. The new orbital period results in Dimorphos completing an extra full orbit roughly every 9.8 days.

The difference between the pre-impact and post-impact mutual orbit period of the Didymos–Dimorphos system greatly exceeds the approximately 7 min period change calculated for the case of a simple momentum transfer with no momentum enhancement[1]. Estimates of

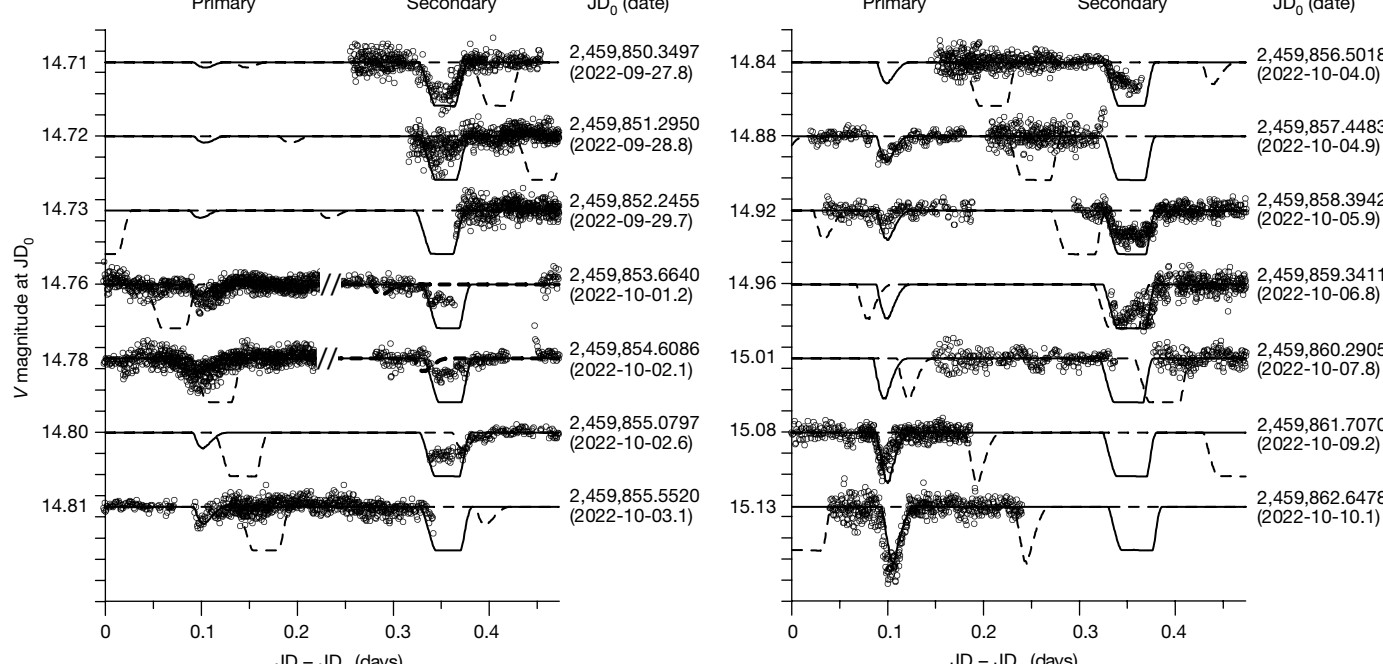

**Fig. 4 | Observed mutual events of the Didymos system.** The data are marked as circles and the solid curve represents the synthetic lightcurve for the best-fit post-impact solution. The dashed curve is the pre-impact orbit prediction from ref. 6. The primary and secondary events are shown on the left and right sides of the plots, respectively. In some cases, the observations of a secondary event precede those of a primary event (that is, their order in the dataset is the inverse of that shown in the plot). We present these events in reverse order and they are separated by a '///' symbol in the plot (0.4728 days are to be subtracted from the x coordinate of data points to the right from this separator). The y axis shows the magnitude on the night of the observation for each dataset and each tick mark has a range of 0.02 magnitudes.

the change in orbital velocity imparted to Dimorphos require modelling beyond the scope of this paper, but it is evident that the ejecta from the DART impact carried a substantial amount of momentum compared with what the DART spacecraft itself was carrying (for example, ref. 21). To serve as a proof-of-concept for the kinetic impactor technique of planetary defence, DART needed to demonstrate that an asteroid could be targeted during a high-speed encounter[16] and that the orbit of the target could be changed. DART has successfully done both.

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

## Methods

The models incorporated three types of observation of the Didymos–Dimorphos system: photometric lightcurves, radar, and Didymos-relative astrometry from the DRACO camera aboard DART[20]. We determined the post-impact orbital period using two separate models (ref. 6, hereafter SP22, and refs. 5,11, hereafter N22+). Both approaches use the same sets of pre-impact and post-impact lightcurves (Extended Data Tables 1 and 2). The SP22 approach models the lightcurve observations to determine the properties of the orbit. The N22+ approach incorporates Didymos-relative astrometry from DRACO optical navigation images to revise the orbital parameters of the pre-impact orbit and includes lightcurve mutual event timings and radar observations for the post-impact solution (Extended Data Tables 3–7).

### Photometric lightcurve data and reductions

Previous observations of the Didymos system[4] demonstrated the need for requirements on the photometry used in the analysis. We define our data quality requirement as an RMS < 0.01 magnitudes, in which the RMS value refers to the consistency over the nightly run and results in a minimum SNR on the individual exposures of about 100. For an accurate decomposition of the lightcurve, we require adequate coverage of the primary lightcurve outside mutual events. We prefer two complete rotation periods of the primary ($P_{rot}$ = 2.26 h) outside the events and estimate this requirement as 6 h of continuous observation. The observations can be split between multiple stations. Four observatories contributed data that met the photometric requirements of the lightcurve dataset for the orbital period change (Extended Data Table 2): Las Campanas Observatory 1-m Swope Telescope, the Las Cumbres Observatory Global Telescope (LCOGT) network 1-m telescopes, the Danish 1.54-m telescope at the European Southern Observatory's La Silla site and the Lowell Observatory 1.1-m Hall telescope.

The Las Campanas Observatory 1-m Swope telescope is located in the Atacama Desert, Chile[22]. The Swope 4K CCD is a visible-wavelength, direct-imaging CCD with a 29.7 × 29.8 arcmin field of view. Swope observations were taken in the Sloan-r' filter and used sidereal tracking with one or two sky pointings each night. Instrumental aperture photometry was performed on every frame using the Python package SEP[23]. We use the astroquery Python package to query VizieR[24] and Horizons[25] databases to identify Gaia stars and to obtain the coordinates of the asteroid for the given date of the images, respectively, and the GaiaXPy Python package, to request and download synthetic photometry of Gaia stars[26] in Sloan-r band when available. The Swope data show discrepancies in the photometry (as seen in Fig. 2) at the 0.01–0.02-magnitude level. There are no issues on the timing of the events, which are the key drivers for the derivation of the new orbit period. Additional reductions of these data with optimized apertures will be used to address these discrepancies.

The LCOGT network[27] consists of telescopes at seven sites around the world, operated robotically using dynamical scheduling software[28]. We used the 1-m telescopes at the South Africa and Chile nodes with the telescopes tracking at half of the ephemeris rates. These observations were scheduled and reduced using the NEOexchange Target and Observation Manager and data-reduction pipeline[29]. Images were pre-processed using the Python-based BANZAI pipeline[30]. Astrometry and photometry was performed using the Python-based NEOexchange pipeline[29]. The LCOGT data were primarily obtained in PanSTARRS-$w$ band (equivalent to a broad $g + r + i$ band) and was calibrated to the Gaia DR2 (ref. 31) using calviacat[32], with the $w$ band treated as an $r$ band. Calibration stars were constrained to have 'solar-like' colours.

The Danish 1.54-m telescope is located at the European Southern Observatory's La Silla site in Chile. Observations were performed by the MiNDSTEp (Microlensing Network for the Detection of Small Terrestrial Exoplanets) consortium. The Danish Faint Object Spectrograph and Camera (DFOSC) instrument, with a field of view 13.7′ × 13.7′, was used in imaging mode. Images were taken with the Bessell R filter using sidereal tracking. Data reduction used a custom Python pipeline, including alignment of frames using Astrometry.net tools[33]. Relative photometry was calibrated using the procedure outlined in ref. 34 using the calviacat[32] package and the Gaia DR3 star catalogue, with conversion to SDSS-r band magnitudes assuming a colour of $(g − r)$ = 0.52 for Didymos[4,35].

The Lowell 1.1-m Hall telescope, located on Anderson Mesa south of Flagstaff, Arizona, is equipped with a 4K × 4K CCD that images a 25-arcmin square field. The telescope was tracked at half of the ephemeris rate. Exposures were taken with a broad $VR$-band filter. Photometric calibration was based on field star magnitudes from the PanSTARRS catalogue. Only stars with high SNR (>100) and solar-like colours were used for calibration. For the 2 October 2022 data, the photometry was measured using the Canopus software package. For the 5 October 2022, the photometry was measured using the PHOTOMETRYPIPELINE[36].

We added lightcurve observations from three telescopes (Extended Data Table 1) to augment the pre-impact lightcurve solutions published in ref. 6 and ref. 5: the 6.5-m Magellan Baade telescope, the SOAR (Southern Astrophysical Research) 4.1-m telescope and the 4.3-m Lowell Discovery Telescope. Both of the revised models confirmed the previous solutions.

### Lightcurve decomposition

To model the photometric data of the binary asteroid system, we follow the decomposition methods defined in refs. 17,37 and discussed in ref. 4. Outside mutual events, the largest signal in the Didymos system lightcurve is the flux of the primary, which can be represented by the following Fourier series:

$$F_1(t) = C_1 + \sum_{k=1}^{m_1} \left[ C_{1k}\cos\frac{2\pi k}{P_1}(t - t_0) + S_{1k}\sin\frac{2\pi k}{P_1}(t - t_0) \right]$$

$F_1(t)$ is the flux of the primary, Didymos, at time $t$, $C_1$ is the mean flux of the primary, $C_{1k}$ and $S_{1k}$ are the Fourier coefficients, $P_1$ is the lightcurve rotational period of Didymos, $t_0$ is the zero-point time and $m_1$ is the maximum significant order. By using this mathematical representation for the system, we assume that Didymos is in principal-axis rotation, that mutual illumination between the objects is negligible and that the rotational lightcurve does not change with time. The lightcurve data are corrected to constant geocentric and heliocentric distances and a consistent solar phase angle. We connect data from different telescopes by scaling them in relative magnitude compared with each other, which has no impact on the timing of the mutual events.

We use observations taken outside mutual events to fit the rotational lightcurve of Didymos. The rapidly changing Earth–Didymos–Sun geometry during this period of Didymos' close approach to Earth causes observable changes in the primary rotational lightcurve. For our previous work[4], we were able to combine data on the timescales of days to weeks. For this dataset, separate decompositions are done for each Julian Day (JD). We correct for the overall fading of the ejecta for each dataset by fitting a linear flux trend before performing the lightcurve decomposition.

### Radar observations

We observed Didymos and Dimorphos using the Goldstone X-band radar (3.5 cm, 8,560 MHz) on the 70-m DSS-14 telescope on 14 dates between 27 September and 13 October 2022. On 2, 6 and 9 October, we also used the 100-m Green Bank Telescope to receive radar echoes in a bistatic configuration with transmissions from Goldstone. Typical transmitter power was 430 kW. We obtained echo power spectra during each of the observing windows and range-Doppler images on several days centred on 4 October when the SNRs were the highest. Didymos was clearly detected in all of the data (>3$\sigma$) and its maximum bandwidth

varied from 22 Hz on 27 September, when the subradar latitude was −50°, to 34 Hz on 13 October, when its subradar latitude was −32° (based on the pole direction estimated by ref. 11).

Detecting Dimorphos was challenging and required experimenting with setups having different frequency resolutions, range resolutions and integration times. This process was a trade-off between obtaining longer integrations with sufficiently high SNRs to detect Dimorphos versus reducing the smearing caused by the orbital motion during the integration. We found that the echo from Dimorphos was most consistently visible at resolutions of 1 Hz in the echo power spectra and at 0.5 Hz in the images. Because of the 11.9-h rotation period, a diameter of about 160 m and a subradar latitude of −50° to −30° (ref. 11), the echo from Dimorphos was expected to have a bandwidth of about 1 Hz (ref. 11), so the data do not resolve Dimorphos in frequency but maximize the SNRs by nearly matching the bandwidth. The contribution of self-noise in the echo power spectra is negligible and does not notably affect the SNRs. We attempted imaging with time delay resolutions of 0.5 μs and 1 μs (corresponding to range resolutions of 75 m and 150 m) and found that the 0.5-μs setup yielded more consistent detections. We experimented with summing data spanning a range of time intervals and found that the echo from Dimorphos was not clearly visible in all the data on any given day. It became more difficult to detect Dimorphos after 4 October, as the distance to Didymos increased and the SNRs correspondingly decreased. Figure 2 shows range-Doppler images and Extended Data Fig. 1 shows selected echo power spectra in which the echo from Dimorphos was seen.

We measured the separations between Dimorphos and Didymos in the echo power spectra and range-Doppler images and used these measurements in the estimation of the orbital parameters of Dimorphos relative to Didymos. The separations in Doppler frequency and range between Didymos and Dimorphos relate to the relative velocity and distance along the line of sight of the observer because of their mutual orbit about each other. We used only data in which both Didymos and Dimorphos were clearly visible for making these measurements. The echo power spectra were processed so that hypothetical echoes from the Didymos system barycentre appear at 0 Hz (ref. 38). Because the reflex motion of Didymos about the system barycentre is <10 m (0.08 Hz)[11], we assumed that the Didymos centre of mass (COM) is at 0 Hz, so that the Doppler frequency of Dimorphos represents the relative Doppler shift. The echo from Dimorphos is unresolved, so we assumed that its COM was located in the Doppler bin that contained the strongest spike from the echo from Dimorphos. We assigned uncertainties of ±2 Hz to the Doppler separation measurements to take into account the uncertainties resulting from the frequency resolution of the spectra (1 Hz), the ephemeris errors in the location of the system barycentre (0.24 Hz, 3σ) and the reflex motion of Didymos about the system barycentre (<0.1 Hz). Consequently, the principal source of uncertainty in measurements of the range-Doppler separations are the Doppler frequencies of Dimorphos.

Due to the low SNRs, the COM of Didymos is hard to locate in the range-Doppler images, so we assumed that it is located 375 m (5 range pixels at 75 m per pixel) behind the leading edge, which is the brightest part of the echo and easiest to see. This distance equals the equatorial radius reported from the 3D shape model obtained by ref. 11 and is consistent with preliminary estimates from the DART spacecraft images reported by ref. 16. The echo from Dimorphos extended over one to three range rows and we assumed that its COM is in the trailing row. We assigned uncertainties of 150 m (two range rows) to the range separation measurements. Extended Data Tables 5 and 6 show the range and Doppler frequency of Dimorphos relative to Didymos that were used in the orbit determination. We estimated eight range measurements on 9 October (when reception at Green Bank facilitated detecting echoes from Dimorphos), far more than on any other day, so we inflated their uncertainties by a factor of 3 to mitigate the effects of correlated errors.

## Didymos-relative optical astrometry from DRACO images

We measured the positions of Dimorphos relative to Didymos in 16 DRACO images taken in the minutes before impact on 26 September 2022 between 23:10:58.235 and 23:12:39.336 UTC to use in the orbit estimation process. At the time these measurements were made, no shape models estimated from spacecraft images were available to fit to the partially illuminated figures of the two bodies, so we measured the intersections of the limbs with the relative position vectors. These measurements were differenced to estimate the limb-to-limb positions of Dimorphos relative to Didymos. These positions were mapped from image coordinates into right ascension (RA) and declination (DEC) using the camera model and the GNC (guidance, navigation and control) spacecraft attitude knowledge. Measurement uncertainties of $1.13 \times 10^{-3}$ degrees (3σ) were derived by repeating this process and comparing the different observations. We assumed the equatorial extents of Didymos and Dimorphos to be 425 m and 88 m, respectively, and added an angular distance corresponding to 425 − 88 = 337 m (±20 m (1σ) uncertainty) in the direction of the limb-to-limb separations to estimate the distances between the COMs. Because the measurements covered a very short time span, we de-weighted the uncertainties by 4× (√16) to mitigate effects of correlated measurement errors. We de-weighted the DEC measurements by an extra factor of 2 because they are clearly noisier than the RA measurements. Extended Data Table 7 lists the observations and uncertainties.

## Orbital period determination by means of lightcurves (SP22 method)

The numerical model of the Didymos system in ref. 6 was developed using the techniques described in refs. 39–41. Didymos and Dimorphos are represented by ellipsoids with axial ratios of $a_1/c_1 = b_1/c_1 = 1.37$, $a_2/c_2 = 1.53$ and $b_2/c_2 = 1.50$ (ref. 16). The motion of the two bodies is assumed to be Keplerian. The post-impact system was analysed with no a priori assumption on the new binary orbital period. The lightcurve data from 28 and 29 September showed that parts of the data were attenuated with respect to the rotational lightcurve of the primary. Those sections of the data were iteratively masked until all of the data points in the mutual events were identified and the lightcurve decomposition was complete. The first mutual event (0.03 magnitudes deep) was determined to be a secondary eclipse, as the system geometry predicted very shallow or absent primary events.

We adapted the method from ref. 6 to estimate the uncertainty of the post-impact period. When stepping the period over a suitable interval, we computed normalized $\chi^2$ for each step. We determined its 3σ uncertainty as an interval in which $\chi^2$ is below a certain limit. The adopted limiting $P$-value corresponds to the probability that the $\chi^2$ exceeds a particular value only by chance equal to 0.27%. At each step of the period scanning, the mean anomaly of Dimorphos at the epoch of the impact was also scanned within its 3σ uncertainty interval that was determined by ref. 6 and that we have revised using the extra data taken in July 2022. The SP22 pre-impact period was 11.921478 ± 0.000123 (3σ) h.

The SP22 model determines a post-impact period of 11.372 ± 0.017 (3σ) h, corresponding to an orbit period change of −33.0 ± 1.0 (3σ) min.

## Orbital period determination by means of radar and lightcurves (N22+ method)

The lightcurve analysis method described in ref. 5 is a less complicated approach compared with the methods presented in ref. 6. However, it has the advantage of combining information from different data types, such as radar, relative optical astrometry from DRACO images and lightcurve mutual events. The pre-impact orbital period using the N22+ approach was 11.92148 ± 0.00013 (3σ) h.

Lightcurve decomposition was done independently from the SP22 process and required identifying mutual events. The first identified

post-impact mutual event was on UTC 28 September 2022. We expected that the head-on impact would decrease the orbital period compared with the pre-impact solution and expected an event with a duration of approximately 1 h. To identify the mutual event, we tested a range of orbit periods from 11 to 12 h in time steps of 0.1 h, with a best match of 11.4 h. Subsequent observations helped refine the initial estimate.

For each mutual event, there are four contact times: when the event begins and flux decreases ($T_1$), when flux reaches a minimum ($T_2$), when the flux begins to increase ($T_3$) and when the event ends and the flux returns to the baseline ($T_4$). We use times $T_{1.5}$ and $T_{3.5}$ in the orbit determination. These times are when the flux is at half the total reduction in flux during the event (Fig. 1 in ref. 5). We use $1\sigma$ uncertainties of $(T_{1.5} - T_1)/2$ and $(T_4 - T_{3.5})/2$ for $T_{1.5}$ and $T_{3.5}$, respectively.

We used a least-squares approach, as described in ref. 5, for estimating the orbital parameters of Dimorphos relative to Didymos. Before the DART impact, Dimorphos is assumed to be a point mass on a modified Keplerian orbit around Didymos, with an extra term for modelling the drift in mean motion from nongravitational effects, such as the binary YORP effect and tidal dissipation. The post-impact orbit was assumed to be Keplerian, as the data-arc length is too short to detect a drift in mean motion. We used $\Delta n$ to capture the change in mean motion owing to the DART impact. The mean anomaly, $M$, and mean motion, $n$, of Dimorphos at time, $t$, are given by:

$$M(t) = M_0 + n_0(t - t_0) + \frac{1}{2}\dot{n}(t - t_0)^2 \text{ for } t < t_{\text{imp}}$$
$$M(t) = M_{\text{imp}} + (n_{\text{imp}} + \Delta n)(t - t_{\text{imp}}) \text{ for } t > t_{\text{imp}}$$
$$n(t) = n_0 + \dot{n}(t - t_0) \text{ for } t < t_{\text{imp}}$$
$$n(t) = n_{\text{imp}} + \Delta n \text{ for } t > t_{\text{imp}}$$

in which $t_{\text{imp}}$ is the time of the DART impact, $M_0$ and $n_0$ are the mean anomaly and mean motion at $t_0$, respectively, $\dot{n}$ is the linear drift in mean motion from nongravitational effects and $M_{\text{imp}}$ and $n_{\text{imp}}$ are the mean anomaly and mean motion at impact, respectively.

We used differential corrections as described in ref. 5 for estimating the orbital parameters $M_0$, $n_0$, $\dot{n}$, $\Delta n$, the pre-impact semimajor axis ($a$) and the orbit pole longitude ($\lambda$) and latitude ($\beta$). This requires calculating a computed value corresponding to each observation using a model. We used three kinds of observations: lightcurve mutual event times, radar range and Doppler measurements of Dimorphos relative to Didymos and the separation of Dimorphos from Didymos as seen in spatially resolved DRACO images. The modelling of the first two observables is described in ref. 5. To model the separation of Dimorphos from Didymos in DRACO images, we used SPICE[42] to subtract the RA and DEC of the COM of Didymos from those of the COM of Dimorphos as seen from the DART spacecraft.

The N22+ approach results in a post-impact period of $11.371 \pm 0.016$ ($3\sigma$) h and an orbit period change of $-33.0 \pm 1.0$ ($3\sigma$) min. The best-fit orbit parameters are presented in Extended Data Table 3.

## Data availability

The lightcurves and radar data used in this analysis of the orbital period are available in the JHU/APL Data Archive at https://lib.jhuapl.edu/papers/orbital-period-change-of-dimorphos-due-to-the-dart/. The DRACO images can be found in an archive associated with the Daly et al. paper (https://lib.jhuapl.edu/papers/dart-an-autonomous-kinetic-impact-into-a-near-eart/). Furthermore, all observations from Las Campanas Observatory, Las Cumbres Observatory Global Telescope (LCOGT) network and the Lowell Discovery Telescope will be publicly archived at the Planetary Data System Small Bodies Node with the DART mission data by October 2023. The radar datasets will be separately archived at the Planetary Data System.

## Code availability

The algorithms used here were published in Scheirich and Pravec[6] and Naidu et al.[5].

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

**Acknowledgements** This work was supported by the DART mission, NASA Contract No. 80MSFC20D0004. Part of this research was carried out at the Jet Propulsion Laboratory, California Institute of Technology, under a contract with the National Aeronautics and Space Administration. The work by P. Scheirich and P.P. was supported by the Grant Agency of the Czech Republic, grant 20-04431S. They appreciate access to computing and storage facilities owned by parties and projects contributing to the National Grid Infrastructure MetaCentrum provided under the programme 'Projects of Large Research, Development, and Innovations Infrastructures' (CESNET LM2015042) and the CERIT Scientific Cloud LM2015085. The Green Bank Observatory is a facility of the National Science Foundation operated under cooperative agreement by Associated Universities, Inc. Observations at the Danish 1.54-m telescope were supported, in part, by the European Union H2020-SPACE-2018-2020 research and innovation programme under grant agreement no. 870403 (NEOROCKS). This work makes use of observations from the Las Cumbres Observatory Global Telescope network. This paper includes data gathered with the 6.5-m Magellan Telescopes located at Las Campanas Observatory, Chile, and is based on observations obtained at the Southern Astrophysical Research (SOAR) telescope, which is a joint project of the Ministério da Ciência, Tecnologia e Inovações do Brasil (MCTI/LNA), the US National Science Foundation's NOIRLab, the University of North Carolina at Chapel Hill (UNC) and Michigan State University (MSU). These results made use of the Lowell Discovery Telescope (LDT) at Lowell Observatory. Lowell is a private, nonprofit institution dedicated to astrophysical research and public appreciation of astronomy and operates the LDT in partnership with Boston University, the University of Maryland, the University of Toledo, Northern Arizona University and Yale University. U.G.J. acknowledges funding from the Novo Nordisk Foundation Interdisciplinary Synergy Programme grant no. NNF19OC0057374 and from the European Union H2020-MSCA-ITN-2019 grant no. 860470 (CHAMELEON). E.K. is supported by the National Research Foundation of Korea 2021M3F7A1082056. P.L.-P. was partly funded by 'Programa de Iniciación en Investigación-Universidad de Antofagasta, INI-17-03'.

**Author contributions** C.A.T. is the lead of the DART mission's Observations Working Group. She coordinated observations, led the paper writing and participated in the observing. S.P.N. and P. Scheirich performed the independent modelling efforts to determine the post-impact period and period change. N.A.M. and P.P. accepted lightcurve data based on the requirements and

performed the decompositions. S.R.C. supported orbit estimation of Dimorphos. A.S.R., N.L.C. and A.F.C. lead the DART Investigation Team, contributed to the writing and revision of this paper, and coordinated inputs across the DART Investigation Team. D.J.O., C.C. and N.M. planned, executed and reduced the data from Las Campanas Observatory's Swope telescope. T.A.L. led the data collection from the Las Cumbres Observatory Global Telescope network. L.A.M.B. and M. Brozović assisted with proposal writing, observing and data processing for the radar observations. A.R., P.K. and K.H. performed data reduction and photometry for the Danish telescope dataset. D.M. and B.R. measured the positions in the OPNAV images. P.A.T., A.D.S., P. Salas, W.P.A. and G.W. helped with the Green Bank Observatory observations. M.W.B. assisted with the proposal for Green Bank Observatory. C.S., U.G.J. and M.D. planned and coordinated observations at the Danish telescope. B.S. and T.P. performed the observations and reductions for the Lowell Hall telescope. M.M.K., T.L.F. and C.E.H. performed the observations for the Lowell Discovery Telescope. J.D.G. assisted with the planning of the radar observations. J.B. generated the spacecraft SPK that was used in the OPNAV treatment. J.C., E.G., S.G. and L.P. built the portal and pipeline for scheduling and reducing the LCOGT data. M. Bonavita, M.J.B., E.K., P.L.-P., M.R. and S.S. performed observations at the Danish Telescope. W.H.R. and E.V.R. assisted with planning the observing effort. H.F.A. provided comments on the manuscript and performed the formatting.

**Competing interests** The authors declare no competing interests.

**Additional information**
**Correspondence and requests for materials** should be addressed to Cristina A. Thomas.

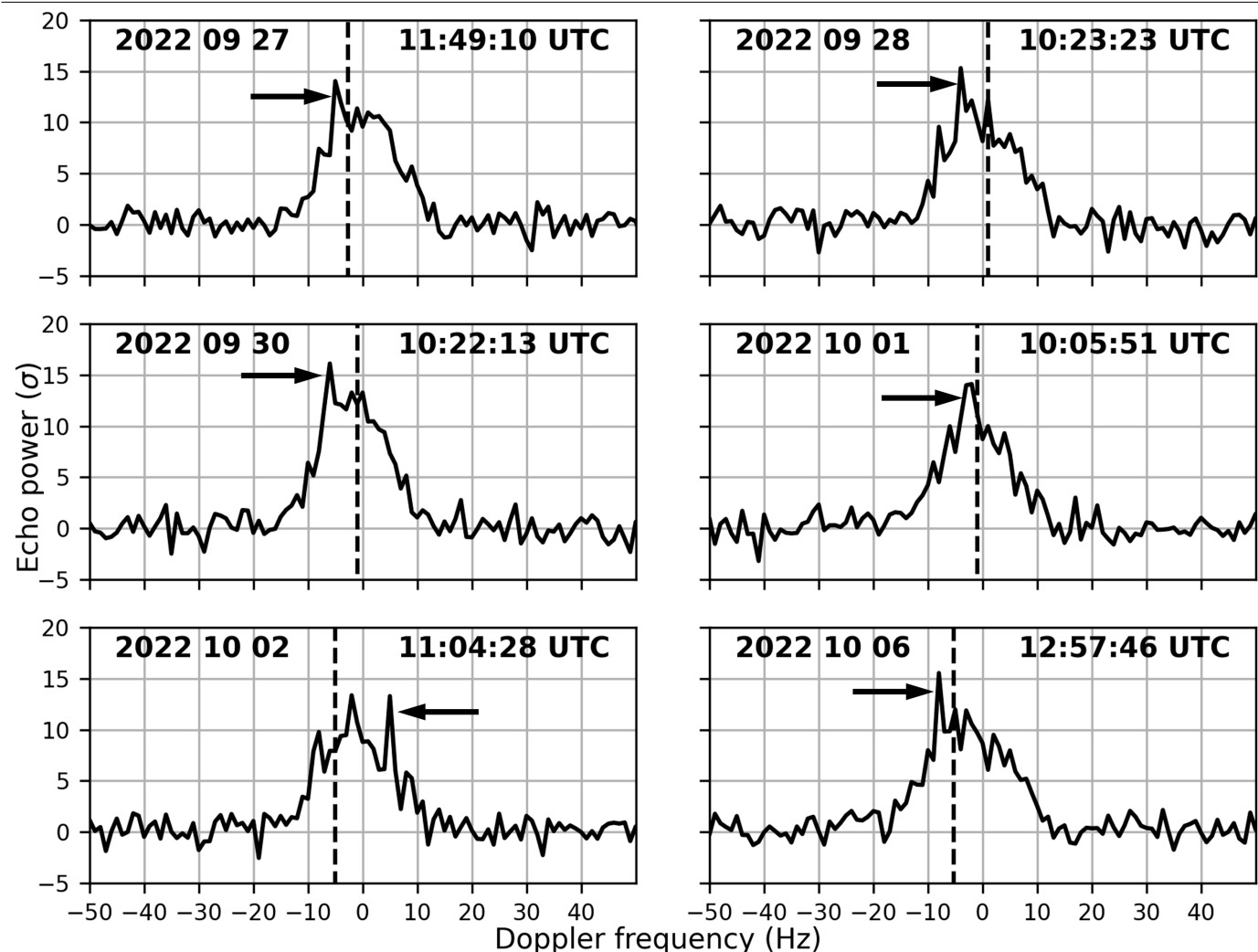

**Extended Data Fig. 1 | Goldstone radar echo power spectra.** Selected radar echo power spectra obtained at Goldstone that were used to measure the Doppler separations in Extended Data Table 6. The spectra were obtained in the opposite sense of circular polarization as the transmitted wave. Each spectrum was integrated for 10–15 min to detect Dimorphos with minimum smear owing to orbital motion (<8°). Echoes from Didymos are centred on 0 Hz and have a bandwidth of between 22 and 34 Hz. The echo from Dimorphos appears as a narrow spike superimposed on the signal from Didymos, a pattern observed with radar observations of dozens of other near-Earth asteroids (for example, ref. 43), indicated by the arrows. The Doppler frequency of Dimorphos varies with time between positive and negative values because of its orbital motion and estimated values can be found in Extended Data Table 6. Dashed vertical lines show the Doppler frequencies of Dimorphos predicted by the pre-impact orbit. Prediction uncertainties are smaller than the resolution of the spectra.

**Extended Data Table 1 | Pre-impact photometric observations**

| Date (UTC) | Start Time (UTC) | Duration (Hr) | # of Points | Telescope | RMS Residual (N22+) mag | RMS Residual (SP22) mag |
|---|---|---|---|---|---|---|
| 2022-07-02 | 03:59 | 6.6 | 193 | LCO/Magellan Baade 6.5-m | 0.008 | 0.009 |
| 2022-07-04 | 06:52 | 3.8 | 129 | CTIO/SOAR 4.1-m | 0.007 | 0.006 |
| 2022-07-05 | 04:24 | 6.4 | 210 | CTIO/SOAR 4.1-m | 0.008 | 0.006 |
| 2022-07-06 | 08:02 | 3.2 | 89 | Lowell Discovery Telescope 4.3-m | 0.005 | 0.006 |
| 2022-07-07 | 07:50 | 3.5 | 85 | Lowell Discovery Telescope 4.3-m | 0.009 | 0.006 |

Pre-impact photometric observations of (65803) Didymos beyond those described by Pravec et al.[4].

**Extended Data Table 2 | Post-impact photometric observations**

| Date (UTC) | Start Time (UTC) | End Time (UTC) | Duration (Hr) | # of Points | Telescope | RMS Residual (N22+) mag | Slope Correction (N22+) mag/day | RMS Residual (SP22) mag | Slope Correction (SP22) mag/day |
|---|---|---|---|---|---|---|---|---|---|
| 2022-09-28 | 2:33 | 6:09 | 3.6 | 237 | LCO/Swope 1-m | 0.008 | 0.10 | 0.008 | 0.07 |
| 2022-09-28 | 2:38 | 9:16 | 6.7 | 340 | La Silla/Danish 1.54-m | 0.006 | 0.10 | 0.006 | 0.12 |
| 2022-09-29 | 2:40 | 9:17 | 6.6 | 433 | LCO/Swope 1-m | 0.007 | 0.24 | 0.007 | 0.25 |
| 2022-09-29 | 2:50 | 9:33 | 6.7 | 639 | La Silla/Danish 1.54-m | 0.005 | 0.10 | 0.005 | 0.12 |
| 2022-09-29 | 4:52 | 8:47 | 3.9 | 212 | CTIO/LCOGT-LSC 1-m | 0.003 | 0.12 | 0.003 | 0.12 |
| 2022-09-30 | 2:40 | 9:41 | 7.0 | 669 | La Silla/Danish 1.54-m | 0.006 | 0.10 | 0.006 | 0.11 |
| 2022-09-30 | 2:52 | 9:15 | 6.4 | 420 | LCO/Swope 1-m | 0.008 | 0.20 | 0.008 | 0.20 |
| 2022-09-30 | 3:52 | 9:16 | 5.4 | 319 | CTIO/LCOGT-LSC 1-m | 0.004 | 0.12 | 0.004 | 0.12 |
| 2022-09-30 | 21:45 | 1:12 | 3.5 | 168 | SAAO/LCOGT-CPT 1-m | 0.004 | 0.22 | 0.004 | 0.21 |
| 2022-10-01 | 3:28 | 9:14 | 5.8 | 376 | LCO/Swope 1-m | 0.007 | 0.22 | 0.006 | 0.23 |
| 2022-10-01 | 4:00 | 9:11 | 5.2 | 292 | CTIO/LCOGT-LSC 1-m | 0.005 | 0.06 | 0.004 | 0.08 |
| 2022-10-01 | 6:33 | 9:28 | 2.9 | 278 | La Silla/Danish 1.54-m | 0.005 | 0.25 | 0.005 | 0.10 |
| 2022-10-01 | 21:48 | 3:09 | 5.4 | 268 | SAAO/LCOGT-CPT 1-m | 0.005 | 0.08 | 0.005 | 0.06 |
| 2022-10-02 | 3:05 | 8:33 | 5.5 | 530 | La Silla/Danish 1.54-m | 0.006 | 0.00 | 0.005 | 0.09 |
| 2022-10-02 | 3:15 | 9:19 | 6.1 | 359 | LCO/Swope 1-m | 0.007 | 0.15 | 0.006 | 0.16 |
| 2022-10-02 | 4:00 | 9:11 | 5.2 | 269 | CTIO/LCOGT-LSC 1-m | 0.004 | 0.02 | 0.003 | 0.00 |
| 2022-10-02 | 8:19 | 10:55 | 2.6 | 132 | Lowell/Hall 1.1-m | 0.005 | 0.15 | 0.006 | 0.09 |
| 2022-10-02 | 22:00 | 00:56 | 2.9 | 136 | SAAO/LCOGT-CPT 1-m | 0.003 | 0.00 | 0.003 | 0.00 |
| 2022-10-03 | 1:00 | 3:02 | 2.0 | 99 | SAAO/LCOGT-CPT 1-m | 0.003 | 0.10 | 0.003 | 0.13 |
| 2022-10-03 | 3:29 | 9:27 | 6.0 | 385 | LCO/Swope 1-m | 0.007 | 0.12 | 0.007 | 0.12 |
| 2022-10-03 | 4:05 | 8:42 | 4.6 | 248 | CTIO/LCOGT-LSC 1-m | 0.004 | 0.12 | 0.004 | 0.10 |
| 2022-10-03 | 4:27 | 5:28 | 1.0 | 98 | La Silla/Danish 1.54-m | 0.006 | 0.20 | 0.005 | 0.12 |
| 2022-10-04 | 3:46 | 8:24 | 4.6 | 224 | LCO/Swope 1-m | 0.006 | 0.15 | 0.006 | 0.11 |
| 2022-10-04 | 4:15 | 8:47 | 4.5 | 248 | CTIO/LCOGT-LSC 1-m | 0.005 | 0.05 | 0.005 | 0.04 |
| 2022-10-04 | 23:15 | 3:04 | 3.8 | 206 | SAAO/LCOGT-CPT 1-m | 0.005 | 0.00 | 0.004 | 0.00 |
| 2022-10-05 | 3:40 | 5:51 | 2.2 | 151 | La Silla/Danish 1.54-m | 0.006 | 0.10 | 0.006 | 0.00 |
| 2022-10-05 | 3:45 | 6:39 | 2.9 | 181 | LCO/Swope 1-m | 0.009 | 0.25 | 0.008 | 0.28 |
| 2022-10-05 | 22:18 | 1:59 | 3.7 | 194 | SAAO/LCOGT-CPT 1-m | 0.006 | 0.04 | 0.005 | 0.00 |
| 2022-10-06 | 3:49 | 9:20 | 5.5 | 346 | LCO/Swope 1-m | 0.010 | 0.06 | 0.009 | 0.06 |
| 2022-10-06 | 4:31 | 8:09 | 3.6 | 194 | CTIO/LCOGT-LSC 1-m | 0.005 | 0.10 | 0.004 | 0.10 |
| 2022-10-06 | 5:03 | 9:27 | 4.4 | 370 | La Silla/Danish 1.54-m | 0.005 | 0.00 | 0.005 | 0.14 |
| 2022-10-07 | 4:10 | 8:28 | 4.3 | 379 | La Silla/Danish 1.54-m | 0.006 | 0.12 | 0.006 | 0.11 |
| 2022-10-07 | 4:52 | 9:08 | 4.3 | 219 | CTIO/LCOGT-LSC 1-m | 0.007 | 0.10 | 0.007 | 0.09 |
| 2022-10-07 | 22:31 | 3:03 | 4.5 | 246 | SAAO/LCOGT-CPT 1-m | 0.007 | 0.05 | 0.006 | 0.05 |
| 2022-10-08 | 4:00 | 9:29 | 5.5 | 471 | La Silla/Danish 1.54-m | 0.007 | 0.15 | 0.006 | 0.06 |
| 2022-10-08 | 22:37 | 3:02 | 4.4 | 200 | SAAO/LCOGT-CPT 1-m | 0.010 | 0.05 | 0.011 | 0.00 |
| 2022-10-09 | 4:49 | 9:12 | 4.4 | 244 | CTIO/LCOGT-LSC 1-m | 0.006 | 0.00 | 0.005 | 0.00 |
| 2022-10-09 | 6:41 | 9:26 | 2.8 | 244 | La Silla/Danish 1.54-m | 0.007 | 0.00 | 0.005 | 0.06 |
| 2022-10-10 | 4:30 | 9:18 | 4.8 | 419 | La Silla/Danish 1.54-m | 0.006 | 0.12 | 0.006 | 0.06 |

Post-impact photometric observations of (65803) Didymos used to derive the new orbital period and period change as a result of impact.

**Extended Data Table 3 | Best-fit orbit parameters using the N22 method**

| Parameter | Estimate +/- $1\sigma$ uncertainties |
|---|---|
| Epoch (UTC) | 2022 Sep 26 23:14:24.183 |
| Orbit pole longitude ($\lambda$, degrees) | 313.3 +/- 5.2 |
| Orbit pole latitude ($\beta$, degrees) | -79.3 +/- 1.0 |
| Pre-impact semimajor axis ($a$, km) | 1.206 +/- 0.035 |
| Mean anomaly at epoch ($M_0$, degrees) | 178.9 +/- 5.5 |
| Pre-impact period (h) | 11.921473 +/- 0.000044 |
| Mean motion at epoch ($n_0$, rad/sec) | $(1.4640214 +/- 0.0000054) \times 10^{-4}$ |
| Rate of change of mean motion ($\dot{n}$, rad/sec$^2$) | $(5.4 +/- 1.6) \times 10^{-18}$ |
| Post-impact period (h) | 11.3712 +/- 0.0055 |
| Period change (min) | -33.02 +/- 0.33 |
| Change in mean motion ($\Delta n$, rad/sec) | $(7.085 +/- 0.070) \times 10^{-6}$ |

The input data are listed in Extended Data Tables 4–7. Note: formal uncertainties are scaled by a factor of 2 to capture errors from unmodelled sources.

**Extended Data Table 4 | Mutual event times measured in post-impact lightcurves for the N22+ approach**

| Time (UTC) | Event type | Unc. (days) | Residuals (sigma) | $\Delta t_{impact}$ (days) |
|---|---|---|---|---|
| 2022 SEP 28 04:28:07 | Beginning of secondary eclipse | 0.011 | 0.50 | 1.22 |
| 2022 SEP 28 05:14:03 | End of secondary eclipse | 0.008 | -0.94 | 1.25 |
| 2022 SEP 29 03:02:00 | Beginning of secondary eclipse | 0.01 | -0.04 | 2.16 |
| 2022 SEP 29 03:39:53 | End of secondary eclipse | 0.0135 | -1.45 | 2.18 |
| 2022 OCT 01 06:11:57 | Beginning of primary eclipse | 0.0075 | 0.58 | 4.29 |
| 2022 OCT 01 06:45:04 | End of primary eclipse | 0.008 | 0.71 | 4.31 |
| 2022 OCT 01 23:12:54 | Beginning of secondary eclipse | 0.0115 | 0.19 | 5.00 |
| 2022 OCT 02 00:08:03 | End of secondary eclipse | 0.0115 | -0.55 | 5.04 |
| 2022 OCT 02 04:43:58 | Beginning of primary eclipse | 0.0075 | -0.36 | 5.19 |
| 2022 OCT 02 05:28:45 | End of primary eclipse | 0.01 | 0.63 | 5.26 |
| 2022 OCT 02 22:57:47 | End of secondary eclipse | 0.0075 | -0.23 | 5.99 |
| 2022 OCT 04 08:13:37 | Beginning of secondary eclipse | 0.01 | 1.23 | 7.37 |
| 2022 OCT 05 00:40:10 | Beginning of primary eclipse | 0.0085 | -1.18 | 8.06 |
| 2022 OCT 05 01:22:30 | End of primary eclipse | 0.008 | -0.50 | 8.09 |
| 2022 OCT 05 23:39:33 | Beginning of primary eclipse | 0.0045 | 0.40 | 9.02 |
| 2022 OCT 05 23:56:15 | End of primary eclipse | 0.0055 | -1.89 | 9.03 |
| 2022 OCT 06 05:29:36 | Beginning of secondary eclipse | 0.005 | 1.23 | 9.26 |
| 2022 OCT 06 06:32:15 | End of secondary eclipse | 0.0055 | 0.17 | 9.30 |
| 2022 OCT 07 04:55:12 | End of secondary eclipse | 0.01 | -1.30 | 10.24 |
| 2022 OCT 08 08:34:39 | Beginning of primary eclipse | 0.0065 | 1.28 | 11.39 |
| 2022 OCT 08 08:51:21 | End of primary eclipse | 0.007 | -0.71 | 11.40 |
| 2022 OCT 09 07:12:25 | Beginning of primary eclipse | 0.0065 | 0.80 | 12.33 |
| 2022 OCT 09 07:33:35 | End of primary eclipse | 0.0085 | -0.64 | 12.35 |
| 2022 OCT 10 05:54:48 | Beginning of primary eclipse | 0.005 | 1.06 | 13.28 |
| 2022 OCT 10 06:14:58 | End of primary eclipse | 0.0055 | -1.19 | 13.29 |

All times are one-way light-time-corrected to reflect the time of the events at the asteroid, not the times that they were observed from Earth. The beginnings and ends of events correspond to $T_{1.5}$ and $T_{3.5}$. The fourth column shows the post-fit residuals (observed – computed) for the solution in Extended Data Table 3, normalized by the 1σ uncertainty listed in the third column. The fifth column shows the time since impact.

**Extended Data Table 5 | Goldstone radar range measurements of Dimorphos relative to Didymos**

| Receive time (UTC) | Range (m) | Unc. (m) | Residuals (sigma) |
|---|---|---|---|
| 2022 OCT 04 11:32:00 | -825 | 150 | -0.17 |
| 2022 OCT 04 11:55:00 | -900 | 150 | -0.36 |
| 2022 OCT 09 10:28:09 | 828 | 450 | -0.14 |
| 2022 OCT 09 10:38:09 | 965 | 450 | 0.10 |
| 2022 OCT 09 10:48:09 | 942 | 450 | 0.00 |
| 2022 OCT 09 10:57:57 | 896 | 450 | -0.13 |
| 2022 OCT 09 11:37:46 | 908 | 450 | -0.03 |
| 2022 OCT 09 11:46:47 | 896 | 450 | -0.00 |
| 2022 OCT 09 11:56:47 | 896 | 450 | 0.08 |
| 2022 OCT 09 12:05:46 | 862 | 450 | 0.08 |

The fourth column shows the post-fit residuals (observed–computed) for the solution in Extended Data Table 3, normalized by the $1\sigma$ uncertainty listed in the third column.

**Extended Data Table 6 | Goldstone radar Doppler measurements of Dimorphos relative to Didymos**

| Receive time (UTC) | Doppler (Hz) | Unc. (Hz) | Residuals (sigma) |
|---|---|---|---|
| 2022 SEP 27 11:22:02 | -3.00 | 2.00 | 0.12 |
| 2022 SEP 27 11:49:09 | -5.00 | 2.00 | -0.22 |
| 2022 SEP 28 10:23:24 | -4.00 | 2.00 | 0.23 |
| 2022 SEP 30 10:22:13 | -6.00 | 2.00 | -0.32 |
| 2022 OCT 01 10:05:51 | -2.50 | 2.00 | -0.27 |
| 2022 OCT 02 11:04:28 | 5.00 | 2.00 | -0.54 |
| 2022 OCT 04 09:58:15 | 7.00 | 2.00 | -0.02 |
| 2022 OCT 06 12:44:16 | -8.00 | 2.00 | -0.18 |
| 2022 OCT 06 12:57:45 | -8.00 | 2.00 | -0.33 |
| 2022 OCT 12 09:37:43 | 8.00 | 2.00 | -0.33 |
| 2022 OCT 12 10:26:49 | 9.00 | 2.00 | 0.20 |
| 2022 OCT 13 09:44:09 | 7.00 | 2.00 | -0.27 |

The fourth column shows the post-fit residuals (observed–computed) for the solution in Extended Data Table 3, normalized by the $1\sigma$ uncertainty listed in the third column.

**Extended Data Table 7 | Didymos-relative optical astrometry of Dimorphos**

| Time (UTC) | ΔRA (deg) | ΔRA unc. (deg) | ΔRA residual (sigma) | ΔDEC (deg) | ΔDEC unc. (deg) | ΔDEC residual (sigma) |
|---|---|---|---|---|---|---|
| 2022-09-26 23:10:58.235 | -0.0514196 | 0.0038304 | 0.202 | -0.0125218 | 0.0034673 | 0.08 |
| 2022-09-26 23:11:04.975 | -0.0534928 | 0.0039585 | 0.12 | -0.0117131 | 0.0034132 | 0.45 |
| 2022-09-26 23:11:11.715 | -0.055801 | 0.0040724 | 0.016 | -0.0125985 | 0.0034639 | 0.32 |
| 2022-09-26 23:11:18.456 | -0.0576213 | 0.004194 | 0.067 | -0.0134683 | 0.0035253 | 0.22 |
| 2022-09-26 23:11:24.233 | -0.0593477 | 0.0043055 | 0.098 | -0.0143021 | 0.0035859 | 0.11 |
| 2022-09-26 23:11:30.973 | -0.0615916 | 0.0044643 | 0.115 | -0.0140441 | 0.0035711 | 0.35 |
| 2022-09-26 23:11:37.713 | -0.0641259 | 0.0046184 | 0.11 | -0.0150902 | 0.0036472 | 0.23 |
| 2022-09-26 23:11:44.453 | -0.0667637 | 0.0047804 | 0.127 | -0.016572 | 0.003762 | 0.01 |
| 2022-09-26 23:11:51.193 | -0.069919 | 0.0049938 | 0.087 | -0.0157601 | 0.0036996 | 0.43 |
| 2022-09-26 23:11:57.933 | -0.0732115 | 0.005196 | 0.076 | -0.0170424 | 0.0038 | 0.29 |
| 2022-09-26 23:12:04.673 | -0.0766707 | 0.0054239 | 0.093 | -0.0180365 | 0.0038846 | 0.26 |
| 2022-09-26 23:12:11.413 | -0.0803118 | 0.0056719 | 0.141 | -0.019363 | 0.004004 | 0.17 |
| 2022-09-26 23:12:18.153 | -0.0849044 | 0.0059557 | 0.094 | -0.0204015 | 0.0040924 | 0.18 |
| 2022-09-26 23:12:24.893 | -0.0893027 | 0.0062723 | 0.161 | -0.0214126 | 0.0041959 | 0.22 |
| 2022-09-26 23:12:31.633 | -0.0948127 | 0.0066306 | 0.14 | -0.0225141 | 0.0042994 | 0.28 |
| 2022-09-26 23:12:39.336 | -0.102054 | 0.0070766 | 0.105 | -0.0253911 | 0.0045736 | 0.02 |