## [Peer Review File · Nature]

Manuscript Title: Orbital Period Change of Dimorphos Due to the DART Kinetic Impact

Reviewer Comments & Author Rebuttals

Reviewer Reports on the Initial Version:

Referee #1 (Remarks to the Author):

The paper from Thomas et al. entitled “Orbital Period Change of Dimorphos Due to the DART Kinetic Impact” presents the result of the first planetary defense mission that had for goal to impact the moon of the asteroid Didymos in order to test asteroid deflection techniques.

The paper presents the result of the impact and shows that the orbital period of Dimorphos (the moon of Didymos) has been changed by 33 minutes. The authors show good evidence of that modification by analyzing optical and radar observations of Didymos/Dimorphos both prior and after the impact. The authors are using well described methodology and accurately describe the data acquisition and analysis. This result is of high significance for the field of planetary defense and fully deserved publication in Nature. My comments for improvement of the manuscript can be found here below.

Line 69: Reference 4 could be added to the list of references for Didymos system characterization

The paragraph starting on line 94 lacks critical information about how information for the Didymos system was obtained before the DART impact.

First it would be interesting to add a line or two about how the binary system was discovered. It is important to stress that the secondary was discovered simultaneously with optical and radar observations. The radar observations (that are nowhere mentioned here) were critical to first confirm the satellite discovery (lightcurves can definitely discover satellites, but only radar observations can provide 100% direct proof). Secondly, the radar observations, by allowing to resolve the target and provide highly accurate ranging and Doppler information on the system were critical to obtain the parameters of the binary system such as the size of the secondary, its rotation period, and orbital properties. Thus, reference 10 should be cited in this paragraph.

Reference <https://ui.adsabs.harvard.edu/abs/2010DPS...42.1317B/abstract> should be added on line 96 as is the first analysis of the radar observations following the discovery in 2003.

Reference <https://iopscience.iop.org/article/10.1088/0004-6256/143/1/24/meta> should also be considered for the analysis of the mutual orbit.

In general it would help the reader to add the time lapsed between the impact and the epoch of observations when discussing the comparison between the pre-impact and post-impact solutions. The phase shift could also be added to help the reader. With the modification of the rotation period by 33 min with a pre-impact rotation period of 11.92h, the phase shift is one complete rotation every ~10 days as it can be seen on 2020-10-06 where the pre and post impact solutions are almost similar.

The equation on line 414 has cos and sin repeated.

What is the significance of the radar detection both in CW and delay-Doppler. CW spectra can be affected by self noise that can produce artificial peaks that are much larger than the statistical noise based on the signal to noise of the data. How significant are the CW detections while other peaks could be confused with Dimorphos. Using the shape model that was derived previously by Naidu et al. 2020 to fit the data and show the user the expected CW spectrum would help the reader. For the delay-Doppler images, could you provide the SNR of the Dimorphos detection? Some of them look to have SNR lower than 3 or 2, but it is hard to assess that on an image. Providing that information in Table 5 would help the reader assess the significance of each detection and the probability of a false detection due to random noise.

In the radar observations section, a lot of text is spent explaining how the COM is determined for Didymos in both CW and delay-Doppler observations. As a shape model, that was obtained using previous radar observations, is available, wouldn't it be easier to fit the new observations using that shape model? This will provide better estimates of the COM than assuming it to be 5 range pixels behind the leading edge.

Referee #2 (Remarks to the Author):

Summary of key results: This paper reports light-curved based and radar doppler-range analyses of the orbit of the binary asteroid Didymos-Dimorphos, whose secondary (Dimorphos) was impacted by the NASA DART Mission spacecraft earlier this year. The two separate analysis techniques both independently determined that the orbital period of Dimorphos decreased by 33 minutes as a result of the impact. This period decrease is significantly larger than the 7 minute change that would result from a purely accretive fully inelastic impact, and therefore has important ramifications for possible future missions to deflect Earth-threatening asteroids.

Originality and significance: The article presents the top-line scientific outcome of the NASA DART Mission, and is therefore of clear significance. The standard of originality is also very readily met. The DART impact resulted in the first measurable ephemeris modification of a celestial object, and thus opens a novel and interesting chapter in Solar System Dynamics.

Data and Methodology: The paper benefits in a somewhat unique way from being able to apply measurement techniques (radar-ranging analysis and photometric light-curve analysis) that are at once (i) routine, (ii) well established and (iii) carried out at high signal-to-noise to a *fully novel situation*. This is very unlike a typical astronomical observational discovery situation where a new phenomenon emerges at 2.5 to 3-sigma significance and existing methods must be pushed to the limit to achieve a Nature-worthy detection. The validity of the approach and the quality of the data thus appear excellent.

Use of statistics: The period decrease of -33.0153 ± 1.0 (3σ) minutes corresponds to a 90σ measurement of the period change. In this context of massive S/N, the statistics and uncertainty

treatments appear fully adequate and sufficient.

Conclusions: The article takes a very narrow approach of measuring and reporting the period change. The interpretation, in terms of momentum transfer from ejecta, is essentially left out, as stated in the last paragraph: "Estimates of the change in orbital velocity imparted to Dimorphos require modeling beyond the scope of this paper".

Suggestions:

line 87: It's not clear to me what exactly what is meant by "Mutual events (occultations and eclipses) occur when the Earth or Sun are close to the orbital plane of the two asteroids"... If I am understanding the system configuration correctly, as seen from the Sun, the inclination of the asteroid binary plane to the orbital plane is small enough so that both occultations and eclipses occur. The illumination dips from the shadows are thus generally visible from Earth as features in the photometric light curve (As is seen in Figure 1). At other times it seems possible that the illuminated bodies may eclipse themselves as seen from Earth. I think it would thus help to better define what is meant by "close".

In figure 2, I think it would be useful to indicate the actual line-of-sight velocity variation that is spanned by the orbit ellipse in the x-direction and the physical distance scale spanned by the orbit ellipse in the y-direction. Readers outside the field will not be very familiar with Doppler-range plots, and it is frustrating to count pixels in an attempt match the physical extent, the rotation, the orbital geometry and the orbital velocity with the radar returns in the plot. This diagram conforms with the convention for this subfield, but it can definitely be made more explanatory for a general audience.

In line 296, the Swope Telescope primary eclipse depth looks like it disagrees by more than a factor of two from the other two eclipses in the orbit-folded light curve of the bottom panel. "Slightly different" thus seems to be too mild a wording. I think that a bit of explanation of how systematic issues conspire create a factor of two difference is in order.

In a similar vein, for the 2022-10-02.1 mutual event curve of Figure 4, there is a high signal-to-noise feature that roughly matches the pre-impact prediction. An explanation of how that can arise in the data would be useful.

References -- seem appropriate.

Clarity and context: seem appropriate.

Author Rebuttals to Initial Comments:

Response to the Referees:

We thank you both for your carefully considered comments on our manuscript. We have made a number of changes that are included below to address your comments.

Referee #1 (Remarks to the Author):

The paper from Thomas et al. entitled “Orbital Period Change of Dimorphos Due to the DART Kinetic Impact” presents the result of the first planetary defense mission that had for goal to impact the moon of the asteroid Didymos in order to test asteroid deflection techniques.

The paper presents the result of the impact and shows that the orbital period of Dimorphos (the moon of Didymos) has been changed by 33 minutes. The authors show good evidence of that modification by analyzing optical and radar observations of Didymos/Dimorphos both prior and after the impact. The authors are using well described methodology and accurately describe the data acquisition and analysis. This result is of high significance for the field of planetary defense and fully deserved publication in Nature. My comments for improvement of the manuscript can be found here below.

Line 69: Reference 4 could be added to the list of references for Didymos system characterization

>This reference has been added here.

The paragraph starting on line 94 lacks critical information about how information for the Didymos system was obtained before the DART impact.

First it would be interesting to add a line or two about how the binary system was discovered. It is important to stress that the secondary was discovered simultaneously with optical and radar observations. The radar observations (that are nowhere mentioned here) were critical to first confirm the satellite discovery (lightcurves can definitely discover satellites, but only radar observations can provide 100% direct proof). Secondly, the radar observations, by allowing to resolve the target and provide highly accurate ranging and Doppler information on the system were critical to obtain the parameters of the binary system such as the size of the secondary, its rotation period, and orbital properties. Thus, reference 10 should be cited in this paragraph.

Reference <https://ui.adsabs.harvard.edu/abs/2010DPS...42.1317B/abstract> should be added on line 96 as is the first analysis of the radar observations following the discovery in 2003.

Reference <https://iopscience.iop.org/article/10.1088/0004-6256/143/1/24/meta> should also be considered for the analysis of the mutual orbit.

> A more extensive review of past observations of the Didymos system is outside of the scope of this paper, but previous papers from team members have included those details. I added reference 10

(Naidu et al. 2020) to the sentence about discovery of the satellite. Naidu et al. 2022 (ref 5) states that the radar was used to “check for consistency” of the orbit solution, but wasn’t used in the calculation of the orbit properties. We have kept the text the same since this work relies on more recent calculations of the size of the secondary (Daly et al. this issue), the rotation period (Pravec et al. 2022), and the orbital properties (Naidu et al., Scheirich et al. 2022). Our two independent analyses were completed using their own methods as described in the 2022 papers and we rely on those methods for this work, but we have added a citation to the Fang and Margot paper in the first paragraph.

In general it would help the reader to add the time lapsed between the impact and the epoch of observations when discussing the comparison between the pre-impact and post-impact solutions. The phase shift could also be added to help the reader. With the modification of the rotation period by 33 min with a pre-impact rotation period of 11.92h, the phase shift is one complete rotation every ~10 days as it can be seen on 2020-10-06 where the pre and post impact solutions are almost similar.

> In the paragraph discussing the solution of the new orbit (starting line 156), we have added the following text: “The new orbital period results in Dimorphos completing an additional full orbit every ~9.8 days.” For additional context, we have added the time since impact (in days) to Extended Data Table 4.

The equation on line 414 has cos and sin repeated.

> This error has been fixed.

What is the significance of the radar detection both in CW and delay-Doppler.

> Two of the CW detections of Dimorphos (Oct 4 and Oct 13) have a signal of 3.5 times the noise standard deviations (σ), the remaining CW detections are $> 5 \sigma$. The detection on Sept 27 (first post-impact detection) is 8σ . Each of the detections in delay-Doppler images is $> 3 \sigma$, but as explained in one of the responses later, there are other factors making the detection in delay-Doppler more significant. We have noted in the text that all detections are $> 3\sigma$ on line 445 (methods) and noted the 8σ detection of the first post-impact observation on line 121.

CW spectra can be affected by self noise that can produce artificial peaks that are much larger than the statistical noise based on the signal to noise of the data. How significant are the CW detections while other peaks could be confused with Dimorphos.

> Each of the CW spectra used in this paper was a sum of at least 250 looks (statistically independent estimates of the echo). Standard deviation of the self noise in a given frequency bin is about $(\text{signal}/\sqrt{N_{\text{looks}}})$ times the background noise std. dev. (σ). The signal of the primary is

about 10 sigmas, so with $N_{\text{looks}} = 250$, we get that the standard deviation of the self noise is about 60% of the background noise. Since self noise and background noise are added in quadrature, the total noise standard derivation is only ~ 1.18 times the background noise standard deviation. So the contribution of self noise to the total noise is negligible and does not affect the SNRs significantly. We included this last sentence in the methods section at line 458.

Using the shape model that was derived previously by Naidu et al. 2020 to fit the data and show the user the expected CW spectrum would help the reader.

> Due to the observing geometry in 2003, the z-axis of the Didymos shape model in Naidu et al. 2020 was not well constrained. The spin uncertainty of that model is also quite large, so it is not possible to properly orient the shape model to generate synthetic corresponding to the 2020 observations. DART images reveal that the z-axis of Didymos is smaller than expected from the Naidu et al. 2020 model. At the time of the observations and analysis, a detailed 3D model (Daly et al.) of Didymos from DART images was not yet available.

For the delay-Doppler images, could you provide the SNR of the Dimorphos detection? Some of them look to have SNR lower than 3 or 2, but it is hard to assess that on an image. Providing that information in Table 5 would help the reader assess the significance of each detection and the probability of a false detection due to random noise.

> In each of the detections, the signal from Dimorphos is spread over a group of adjacent pixels, with at least 1 pixel having $\text{SNR} > 3$. The signal can also be seen moving from one image to the next and it follows a trajectory consistent with the prediction (yellow ellipses in Figure 2). Considering these factors, the significance of detection is quite high but not appropriately captured by the SNR of the pixel, so we do not think it is appropriate to provide this in the table.

In the radar observations section, a lot of text is spent explaining how the COM is determined for Didymos in both CW and delay-Doppler observations. As a shape model, that was obtained using previous radar observations, is available, wouldn't it be easier to fit the new observations using that shape model? This will provide better estimates of the COM than assuming it to be 5 range pixels behind the leading edge.

> The shape model obtained from the 2003 radar observations has significant uncertainties, especially along the z-dimension which was not clearly seen in 2003. The z-dimension of the radar shape model also differs by more than 100 meters from that observed by the DART spacecraft. We are also not able to orient the model correctly to the 2020 data due to the large spin period uncertainty. The observation geometry in 2022 also differs significantly from 2003, so we think that using the shape model would not provide better estimates of the primary COM.

Referee #2 (Remarks to the Author):

Summary of key results: This paper reports light-curved based and radar doppler-range

analyses of the orbit of the binary asteroid Didymos-Dimorphos, whose secondary (Dimorphos) was impacted by the NASA DART Mission spacecraft earlier this year. The two separate analysis techniques both independently determined that the orbital period of Dimorphos decreased by 33 minutes as a result of the impact. This period decrease is significantly larger than the 7 minute change that would result from a purely accretive fully inelastic impact, and therefore has important ramifications for possible future missions to deflect Earth-threatening asteroids.

Originality and significance: The article presents the top-line scientific outcome of the NASA DART Mission, and is therefore of clear significance. The standard of originality is also very readily met. The DART impact resulted in the first measurable ephemeris modification of a celestial object, and thus opens a novel and interesting chapter in Solar System Dynamics.

Data and Methodology: The paper benefits in a somewhat unique way from being able to apply measurement techniques (radar-ranging analysis and photometric light-curve analysis) that are at once (i) routine, (ii) well established and (iii) carried out at high signal-to-noise to a *fully novel situation*. This is very unlike a typical astronomical observational discovery situation where a new phenomenon emerges at 2.5 to 3-sigma significance and existing methods must be pushed to the limit to achieve a Nature-worthy detection. The validity of the approach and the quality of the data thus appear excellent.

Use of statistics: The period decrease of -33.0153 ± 1.0 ($3\text{-}\sigma$) minutes corresponds to a 90- σ measurement of the period change. In this context of massive S/N, the statistics and uncertainty treatments appear fully adequate and sufficient.

Conclusions: The article takes a very narrow approach of measuring and reporting the period change. The interpretation, in terms of momentum transfer from ejecta, is essentially left out, as stated in the last paragraph: "Estimates of the change in orbital velocity imparted to Dimorphos require modeling beyond the scope of this paper".

Suggestions:

line 87: It's not clear to me what exactly what is meant by "Mutual events (occultations and eclipses) occur when the Earth or Sun are close to the orbital plane of the two asteroids"... If I am understanding the system configuration correctly, as seen from the Sun, the inclination of the asteroid binary plane to the orbital plane is small enough so that both occultations and eclipses occur. The illumination dips from the shadows are thus generally visible from Earth as features in the photometric light curve (As is seen in Figure 1). At other times it seems possible that the illuminated bodies may eclipse themselves as seen from Earth. I think it would thus help to better define what is meant by "close".

> Thanks for this suggestion. We have added the following text to this paragraph to clarify the system geometry: "For the Didymos-Dimorphos system, mutual events occur when the Didymos-Sun or the Didymos-Earth vector forms an angle less than ~ 17 degrees with the mutual orbit plane of the system. Since the inclination of the mutual orbit to the heliocentric orbit of the binary system is lower than

this value, eclipses (mutual shadowing of the components, Figure 1) always occur. Occultations did not occur during the observing period presented in this paper.”

In figure 2, I think it would be useful to indicate the actual line-of-sight velocity variation that is spanned by the orbit ellipse in the x-direction and the physical distance scale spanned by the orbit ellipse in the y-direction. Readers outside the field will not be very familiar with Doppler-range plots, and it is frustrating to count pixels in an attempt match the physical extent, the rotation, the orbital geometry and the orbital velocity with the radar returns in the plot. This diagram conforms with the convention for this subfield, but it can definitely be made more explanatory for a general audience.

> We added the following text to the Figure 2 caption: “On October 4, the ellipse spans -870 m to +870 m along the y-axis and -7 Hz to +7 Hz along the x-axis, corresponding to line of sight velocity of -12 cm/s to +12 cm/s. On October 9, the ellipse spans -980 m to +980 m along the y-axis and -8 Hz to +8 Hz along the x-axis, corresponding to line of sight velocity of -14 cm/s to +14 cm/s. The physical extents of the ellipse vary due to the viewing geometry.”

In line 296, the Swope Telescope primary eclipse depth looks like it disagrees by more than a factor of two from the other two eclipses in the orbit-folded light curve of the bottom panel. "Slightly different" thus seems to be too mild a wording. I think that a bit of explanation of how systematic issues conspire create a factor of two difference is in order.

> The discrepancy of the mutual event depths seen in the Swope October 2 data can be seen in our figure 2, but there are other discrepancies at the ~ 0.01 - 0.02 mag level in the data. Additional reductions of this data with a better optimized aperture will be used to address these issues. This detailed analysis is ongoing and beyond the scope of this particular paper. We note that there are no issues on the timing of the events, which are the key drivers for the derivation of the new orbital period. The new reductions will inform our ongoing work that investigates the post-impact orbital properties of the Didymos-Dimorphos system.

In a similar vein, for the 2022-10-02.1 mutual event curve of Figure 4, there is a high signal-to-noise feature that roughly matches the pre-impact prediction. An explanation of how that can arise in the data would be useful.

> There are two observed mutual events on this date, whose times match the post-impact solution (solid curve). They also roughly match the pre-impact prediction (dashed curve), because the primary event from the pre-impact solution roughly coincides with the secondary event from the post-impact solution, and vice versa, on this date. To enable readers to better compare the two solutions the following text was added to line 161: “The new orbital period results in Dimorphos completing an additional full orbit every ~ 9.8 days.”

References -- seem appropriate.

Clarity and context: seem appropriate.